# Cystitis-Related Bladder Pain Involves ATP-Dependent HMGB1 Release from Macrophages and Its Downstream H_2_S/Ca_v_3.2 Signaling in Mice

**DOI:** 10.3390/cells9081748

**Published:** 2020-07-22

**Authors:** Shiori Hiramoto, Maho Tsubota, Kaoru Yamaguchi, Kyoko Okazaki, Aya Sakaegi, Yuki Toriyama, Junichi Tanaka, Fumiko Sekiguchi, Hiroyasu Ishikura, Hidenori Wake, Masahiro Nishibori, Huy Du Nguyen, Takuya Okada, Naoki Toyooka, Atsufumi Kawabata

**Affiliations:** 1Laboratory of Pharmacology and Pathophysiology, Faculty of Pharmacy, Kindai University (Formerly Known as Kinki University), Higashi-Osaka 577-8502, Japan; sh02.26hs@gmail.com (S.H.); maho@phar.kindai.ac.jp (M.T.); yamaguchi.kfb@om.asahi-kasei.co.jp (K.Y.); 1511610099c@kindai.ac.jp (K.O.); ayasakaegi0811@gmail.com (A.S.); tori07929@yahoo.co.jp (Y.T.); junichi0927@gmail.com (J.T.); fumiko@phar.kindai.ac.jp (F.S.); 2Division of Emergency and Critical Care Medicine, Fukuoka University Hospital, Fukuoka 814-0180, Japan; ishikurah@fukuoka-u.ac.jp; 3Department of Pharmacology, Okayama University Graduate School of Medicine, Dentistry and Pharmaceutical Sciences, Okayama 700-8558, Japan; wake-h@cc.okayama-u.ac.jp (H.W.); mbori@md.okayama-u.ac.jp (M.N.); 4Faculty of Engineering, University of Toyama, Toyama 930-8555, Japan; nhdu@hcmus.edu.vn (H.D.N.); tokada@eng.u-toyama.ac.jp (T.O.); toyooka@eng.u-toyama.ac.jp (N.T.)

**Keywords:** interstitial cystitis/bladder pain syndrome (IC/BPS), cyclophosphamide (CPA), high mobility group box 1 (HMGB1), receptor for advanced glycation end products (RAGE), Ca_v_3.2 T-type Ca^2+^ channel, hydrogen sulfide (H_2_S), cystathionine-γ-lyase (CSE), macrophage, Adenosine triphosphate (ATP), reactive oxygen species (ROS)

## Abstract

Cystitis-related bladder pain involves RAGE activation by HMGB1, and increased Ca_v_3.2 T-type Ca^2+^ channel activity by H_2_S, generated by upregulated cystathionine-γ-lyase (CSE) in mice treated with cyclophosphamide (CPA). We, thus, investigated possible crosstalk between the HMGB1/RAGE and CSE/H_2_S/Ca_v_3.2 pathways in the bladder pain development. Bladder pain (nociceptive behavior/referred hyperalgesia) and immuno-reactive CSE expression in the bladder were determined in CPA-treated female mice. Cell signaling was analyzed in urothelial T24 and macrophage-like RAW264.7 cells. The CPA-induced bladder pain was abolished by pharmacological inhibition of T-type Ca^2+^ channels or CSE, and genetic deletion of Ca_v_3.2. The CPA-induced CSE upregulation, as well as bladder pain was prevented by HMGB1 inactivation, inhibition of HMGB1 release from macrophages, antagonists of RAGE or P2X_4_/P2X_7_ receptors, and N-acetylcysteine, an antioxidant. Acrolein, a metabolite of CPA, triggered ATP release from T24 cells. Adenosine triphosphate (ATP) stimulated cell migration via P2X_7_/P2X_4_, and caused HMGB1 release via P2X_7_ in RAW264.7 cells, which was dependent on p38MAPK/NF-κB signaling and reactive oxygen species (ROS) accumulation. Together, our data suggest that CPA, once metabolized to acrolein, causes urothelial ATP-mediated, redox-dependent HMGB1 release from macrophages, which in turn causes RAGE-mediated CSE upregulation and subsequent H_2_S-targeted Ca_v_3.2-dependent nociceptor excitation, resulting in bladder pain.

## 1. Introduction

Interstitial cystitis/bladder pain syndrome (IC/BPS) is common to women, and characterized by pelvic/suprapubic pain, bladder discomfort or pressure, as well as increased frequency of urination, which are largely resistant to drug therapy. Apart from typical IC with severe urothelial lesions in the bladder, there are many BPS patients who complain of almost the same bladder symptoms, including bladder pain, even without cystoscopic evidence for cystitis [1]. To identify potential therapeutic targets to treat bladder pain in IC/BPS patients, we have been studying the mechanisms underlying bladder pain in a cyclophosphamide (CPA)-induced cystitis mouse model, and have shown the two different mechanisms involved in the cystitis-related bladder pain, i.e., the activation of the receptor for advanced glycation end products (RAGE) by high mobility group box 1 (HMGB1) [2], and the elevated Ca_v_3.2 T-type Ca^2+^ channel activity by H_2_S generated by upregulated cystathionine-γ-lyase (CSE) [3].

HMGB1, a nuclear protein, once released to the extracellular space, plays a role as a damage-associated molecular pattern (DAMP) or alarmin protein, and promotes not only inflammation [4,5], but also pain signals [6]. Extracellular HMGB1 in the peripheral tissue is considered to participate in inflammatory and neuropathic pain through direct or indirect activation of RAGE, Toll-like receptor 4 (TLR4) or CXCR4 [2,7,8,9,10]. We have demonstrated that the activation of RAGE by HMGB1 derived from unknown cells plays a role in the development of the bladder pain accompanying CPA-induced cystitis in mice [2]. On the other hand, the mice treated with CPA show increased protein expression of CSE, an H_2_S-generating enzyme, in the bladder tissue and Ca_v_3.2 T-type Ca^2+^ channels, the activity of which is enhanced by H_2_S, in the primary afferents [3,11]. Pharmacological inhibition of CSE or T-type Ca^2+^ channels and knockdown of Ca_v_3.2 suppress the CPA-induced cystitis-related bladder pain [3,11]. Thus, as does the HMGB1/RAGE pathway, the CSE/H_2_S/Ca_v_3.2 pathway is considered to participate in the bladder pain accompanying CPA-induced cystitis in mice. Nonetheless, the crosstalk or relationship between these two different pathways is still open to question.

We have demonstrated the involvement of NF-κB in the CPA-induced over-expression of CSE in the bladder [12]. Given NF-κB is a major downstream signal of RAGE activation [4,5], in the present study, we wished to understand whether the CSE/H_2_S/Ca_v_3.2 pathway could be downstream of the HMGB1/RAGE pathway, in the development of bladder pain, accompanying the CPA-evoked cystitis. Another issue to be addressed was in identifying the origin of the extracellular HMGB1 that contributes to the CPA-induced bladder pain. We, thus, examined whether macrophage-derived HMGB1 would participate in the CPA-induced CSE upregulation, possibly responsible for the endogenous H_2_S-mediated, Ca_v_3.2-dependent bladder pain, as it did in the acute pancreatitis-related pain and paclitaxel-induced peripheral neuropathy in our previous studies [8,10]. Further, we also analyzed the roles of ATP and reactive oxygen species (ROS) in the CPA-induced CSE upregulation in the bladder, considering the evidence that CPA causes ATP-dependent macrophage infiltration and ROS generation in the bladder [13,14]. In order to verify the validity of Ca_v_3.2 as a therapeutic target for treatment of bladder pain, we evaluated the effect of novel T-type Ca^2+^ channel blockers and genetic deletion of Ca_v_3.2 on the bladder pain in the mice with CPA-induced cystitis.

## 2. Materials and Methods

### 2.1. Animals

Female ddY mice (4–5 weeks old) were purchased from Kiwa Laboratory Animals Co., Ltd. (Wakayama, Japan). Male and female C57BL/6J mice and Ca_v_3.2^-/-^ mice of a C57BL/6J background were obtained from Kiwa Laboratory Animal Co., Ltd., and Jackson Laboratory (Bar Harbor, ME, USA), respectively, and were bred and mated in Kindai University to produce heterozygous Ca_v_3.2^+/-^ mice, which were used to obtain female homozygous Ca_v_3.2^-/-^ and wild-type Ca_v_3.2^+/+^ mice. The animals were housed in a temperature-controlled room under a 12-h day/night cycle at 24 °C and had enough food and water, and the females were used for experiments. All experimental protocols were approved by Kindai University’s Committee for the Care (Approval number; KAPS-27-002) and Use of Laboratory Animals and were in agreement with the Guiding Principles approved by the Japanese Pharmacological Society and with the Guide for the care and Use of laboratory Animals published by the US national Institutes of Health.

### 2.2. Major Chemicals

Cyclophosphamide (CPA), DL-propargylglycine (PPG), sodium 2-mercaptoethanesulfonate (mesna), ethyl pyruvate, ATP, PSB-12062, N-acetylcysteine, pyrrolidine dithiocarbamate (PDTC) and dimethylsulfoxide (DMSO) were purchased from Sigma-Aldrich (St Louis, MO, USA). TTA-A2 was obtained from Alomone Labs. (Jerusalem, Israel), and A438079, 5-BDBD and AZ 10606120 were from Tocris Bioscience (Bristol, UK). (2*R/S*)-6-Prenylnaringenin (6-PNG) and KTt-45 [6-(3-ethylpent-2-enyl)-5,7-dihydroxy-2-(2-hydroxyphenyl)chroman-4-one] were synthesized in-house, as reported previously [15]. An anti-HMGB1-neutralizing antibody and normal rat IgG (control) were made in-house, the specificity of the antibody being described elsewhere [16]. Recombinant human soluble thrombomodulin (thrombomodulin alfa) was a gift from Asahi Kasei Pharma (Tokyo, Japan). FPS-ZM1 and SB203580 were purchased from Calbiochem (San Diego, CA, USA), and liposomal clodronate and the control liposomes were from FormuMax Scientific Inc. (Sunnyvale, CA, USA). 6-PNG and KTt-45 were suspended in 0.5% carboxymethylcellulose (CMC) sodium solution. CPA, ethyl pyruvate, A438079 and AZ 10606120 were dissolved in saline. TTA-A2 was dissolved in a solution containing 1.3% DMSO, 9.9% Tween 80 and 0.5% methylcellurose. PSB-12062 was dissolved in corn oil containing 1.2% DMSO. 5-BDBD was dissolved in DMSO and diluted with saline containing 5% Tween 80 (final concentration of DMSO was 0.85%) for in vivo experiments. The antibody and the control rat IgG were dissolved in 0.01 M phosphate-buffered saline (PBS), and thrombomodulin alfa was in 0.002% Tween 80-containing saline. FPS-ZM1 was dissolved in DMSO and diluted with 10% Tween 80-containing saline (final concentration of DMSO: 0.5%) for in vivo experiments. SB203580 was dissolved in DMSO. N-Acetylcysteine was dissolved in saline for in vivo experiments, and in the culture medium for in vitro experiments. All other chemicals were dissolved in distilled water.

### 2.3. Induction of Cystitis and Related Bladder Pain by CPA, and Assessment of Bladder Pain-Like Nociceptive Behavior, Referred Hyperalgesia and Bladder Swelling in the Mice

As described previously [17,18], mice received i.p. administration of CPA at 400 mg/kg. Bladder pain-like nociceptive behavior was observed and counted for a 30-min period starting 3.5 h after CPA treatment, immediately followed by evaluation of referred hyperalgesia. For the behavioral experiments, the mice were placed on a raised wire-mesh floor covered by a clear transparent plastic box (23.5 × 16.6 × 12.4 cm) and acclimated to the environment at least for 1 h. The number of nociceptive behaviors, such as licking of the skin of the lower abdomen close to the bladder or pressing the lower abdomen against the floor, was measured for 30 min according to the above-mentioned schedule. Immediately after the behavioral observation, hyperalgesia/allodynia was evaluated, as described in detail elsewhere [11]. Briefly, the region between the anus and urethral opening was stimulated with each of four distinct von Frey filaments (0.008, 0.07, 0.4 and 1.0 g) at intervals of 5–10 s, 10 times in total. The scoring of nociceptive behavior was as follows: 0 = no response; 1 = licking or biting of the external urethral opening and/or the surrounding area, leaving the position, bending of the trunk, raising the upper half of the body and thrashing limbs; 2 = jumping. The sum of nociceptive scores in responses to 10 challenges with each filament was calculated (i.e., the maximal score could be 20). After the behavioral experiments, the mice were killed by cervical dislocation and the wet weight of the excised bladder was measured as an indicator of bladder swelling.

### 2.4. Drug Administration Schedule

To evaluate therapeutic effects of T-type Ca^2+^ channel blockers on the established cystitis-bladder pain, 6-PNG at 10–30 mg/kg, KTt-45 at 10–30 mg/kg and TTA-A2 at 1 mg/kg were administered i.p. to mice 3 h 15 min after i.p. CPA. To analyze the molecular mechanisms for the development of bladder pain, PPG, a CSE inhibitor, at 50 mg/kg or ethyl pyruvate capable of inhibiting HMGB1 release from macrophages at 80 mg/kg was administered i.p. to mice 1 h before i.p. CPA. An anti-HMGB1-neutralizing antibody at 1 mg/kg, thrombomodulin alfa capable of inactivating HMGB1 at 10 mg/kg, FPS-ZM1, a RAGE antagonist, at 1 mg/kg, or N-acetylcysteine, an antioxidant, at 100 mg/kg was administered i.p. 30 min before i.p. CPA. A438079 at 17 mg/kg [13] and AZ10606120 at 0.03–1 mg/kg [19], P2X_7_ antagonists, and 5-BDBD at 1.5–3 mg/kg [20] and PSB-12062 at 1–3 mg/kg [21], P2X_4_ antagonists, were administered in the same schedule as the above. Minocycline, an inhibitor of macrophage/microglia, at 30 mg/kg was administered i.p. twice, i.e., 1 h and 24 h before i.p. CPA. Mesna, an acrolein-quenching drug, was administered orally twice, at 80 mg/kg 30 min before, and 160 mg/kg 2 h after i.p. CPA (i.e., a total dose of 240 mg/kg) [22].

### 2.5. Determination of Expression of Cystathionine-γ-Lyase (CSE) Protein in the Bladder of Mice

Protein levels of CSE in the isolated bladder were determined by Western blotting, essentially as described previously [3]. Briefly, the bladder samples were homogenized in a RIPA buffer [PBS, 1% Igepal CA-630, 0.5% sodium deoxycholate and 0.1% sodium dodecyl sulfate (SDS)] containing 0.1 mg/mL PMSF, 0.15 U/mL aprotinin and 1 mM sodium orthovanadate, and analyzed by Western blot analysis using primary antibodies, i.e., anti-CSE rabbit (Sigma-Aldrich Japan) (1:1000 dilution) and anti-GAPDH rabbit (Santa Cruz Biotechnol., Inc., Santa Cruz, CA, USA) (1:5000 dilution) polyclonal antibodies. A horseradish peroxidase (HRP)-conjugated anti-rabbit IgG antibody (Cell Signaling Technol., Beverly, MA, USA) (1:5000 dilution) was used as a secondary antibody, and immunolabeled proteins were visualized with an enhanced chemiluminescence detection regent (Nacalai Tesque, Kyoto, Japan) and detected by Image Quant 400 (GE Healthcare, Little Chalfont, Buckinghamshire, UK). The density of detected bands was quantified by densitometry.

### 2.6. Immunohistochemical Analysis of Localization of CSE in the Bladder of Mice

The mice were anesthetized with i.p. administration of a mixture of midazolam at 4 mg/kg and medetomidine at 0.3 mg/kg, followed by i.p. sodium pentobarbital at 10 mg/kg, and transcardially perfused with 4% paraformaldehyde (PFA) 4 h after i.p. CPA. The bladder was isolated and fixed in paraffin. The paraffin-fixed bladder was sectioned at 7-μm thickness, deparaffinized, washed and incubated with an antigen retrieval solution (DAKO, Glostrup, Denmark) at 121 °C for 15 min. After washing, the sections were immersed in 0.3% hydrogen peroxide solution containing methanol for 20 min, and subjected to blocking with PBS containing 1% normal goat serum (NGS) and 10% Triton X-100 for 30 min. The sections were incubated with an anti-CSE rabbit polyclonal antibody (1:1000; Sigma-Aldrich Japan) at 4 °C overnight. After washing, the sections were incubated with a biotinylated anti-rabbit IgG antibody for 1 h and then reacted with an avidin/biotin complex (VECSTATIN Elite ABC kit, Vector, Burlingame, CA, USA) for 30 min at 4 °C. After immersion with a 0.1 M Tris-HCl buffer (pH 7.6) containing 0.05% 3,3′-diaminobenzidine-tetra HCl and 0.015% hydrogen peroxide for 10 s at 24 °C, the sections were rinsed with the Tris-HCl buffer and water and enclosed by Entellan™ (Merck, Darmstadt, Germany).

### 2.7. Macrophage Depletion

Liposomal clodronate or the control liposome at 1.05 mg/mouse was administered i.p. to mice 24 h before i.p. CPA. In preliminary experiments, to confirm macrophage depletion in the isolated spleen, cells positive to both anti-CD11b (a marker for monocyte, macrophage and microglia) and anti-F4/80 (a marker for macrophage and microglia) antibodies was detected and counted by flow cytometry, as described previously [10].

### 2.8. Accumulation of Macrophages in the Bladder of Mice Treated with CPA

To check macrophage accumulation in the bladder, the mice were anesthetized, as mentioned above, 4 h after i.p. CPA, and transcardially perfused with 20 mL of ice-cold saline followed by 50 mL of 4% PFA in 0.1 M phosphate buffer (PB). The bladder was removed and post-fixed in 4% PFA in 0.1 M PB for 3 days at 4 °C, and then cryoprotected in 30% sucrose in 0.1 M PB at 4 °C. The tissue was fixed in Cryomatrix™ (Thermo Fisher Scientific, Waltham, MA, USA) and frozen. Sections were cut at 7-μm thickness by a cryostat and mounted on slides (Dako, Tokyo, Japan). The sections were dried at 37 °C overnight, and then washed with PBS at 5 times and incubated for 2 h in a blocking solution of 4% bovine serum albumin (BSA) in PBS with 0.1% Triton X-100 at room temperature. After 2-h blocking, the sections were incubated overnight at 4 °C with an anti-mouse F4/80 rat monoclonal antibody (AbD Serotec, Oxford, UK) (1:500 dilution) in 1% BSA in PBS with 0.1% Triton X-100. After washing with PBS at 5 times, the sections were incubated with a Cy3^®^-conjugated goat anti-rat IgG antibody (H + L) (Thermo Fisher Scientific) (1:200 dilution) for 2 h at room temperature. Cell nuclei were counterstained with 4′,6-diamidino-2-phenylindole (DAPI) (Sigma-Aldrich) in TBS buffer. The sections were washed at 3 times, and then mounted by a cover slip with Fluoromount™ aqueous mounting medium. Immunoreactivity was observed using a confocal laser fluorescence microscope (FV10C-O, Olympus, Tokyo, Japan), and the number of F4/80-positive cells were determined in each visual field.

### 2.9. Cell Culture

Mouse macrophage-like RAW264.7 cells were cultured in RPMI-1640 medium (Nacalai Tesque, Inc.) supplemented with 100 U/mL penicillin, 100 μg/mL streptomycin (Nacalai Tesque, Inc.) and 10% fetal calf serum (FCS). Human urinary bladder carcinoma T24 cells were cultured in E-MEM medium (FUJIFILM Wako Pure Chemical Co., Osaka, Japan) supplemented in the same manner.

### 2.10. Determination of ATP Release from T24 Cells

T24 cells (72 × 10^4^ cells/dish) were seeded in 60-mm culture dishes and cultured in the 10% FCS-containing culture medium for 24 h. After additional incubation for 24 h in the FCS-free medium, the cells were stimulated with acrolein at 50 μM for 30 min. Thereafter, the amount of ATP in the collected supernatant was quantified by determination of luminescence generated by a luciferin-luciferase method (CheckLite 250 Plus, Kikkoman Biochemifa Company, Tokyo, Japan) using GloMax 20/20 Luminometer (Promega, Madison, WI, USA).

### 2.11. Determination of HMGB1 Release from RAW264.7 Cells and T24 Cells

RAW264.7 cells (12 × 10^4^ cells/dish) were seeded in a 6-well plate and cultured in 10% FCS-containing culture medium for 24 h. After additional 24-h incubation in 1% FCS-containing culture medium, the cells were stimulated with ATP at 0.1, 0.3 or 1 mM for 1, 3 or 6 h, and then the supernatant was collected. T24 cells (72 × 10^4^ cells/dish) were seeded in a 60-mm culture dishes and cultured in 10% FCS-containing culture medium for 24 h. After incubation in the FCS-free medium for 24 h, the cells were stimulated with acrolein at 50, 100 or 200 μM for 3 h or 24 h, and the supernatant was collected. The protein levels of HMGB1 in the collected supernatant were determined using an ELISA kit (Fuso Industries, Ltd., Osaka, Japan) according to the manufacturer’s instructions, and by Western blotting, as reported elsewhere [8]. In Western blotting, after addition of one-fourth volume of 10% SDS buffer containing 312.5 mM Tris-HCl and 50% glycerol (pH 6.8), the supernatant was denatured at 95–100 °C for 5 min in the presence of 2-mercaptoethanol and bromophenol, and processed as described above. An anti-HMGB1 rabbit polyclonal antibody (Abcam, Cambridge, UK) (1:5000 dilution) as a primary antibody and an HRP-conjugated anti-rabbit IgG antibody (Cell Signaling Technology) (1:5000 dilution) as a secondary antibody were used.

### 2.12. Knockdown of P2X_4_ or P2X_7_ by siRNA

To silence the expression of P2X_4_ and P2X_7_, we used the Silencer^TM^ Select Pre-Designed siRNA for mouse P2X_4_ (ID; s71184, s71185) and P2X_7_ (ID; s71188, s71189) (Ambion, Carlsbad, CA, USA), and also employed the negative control siRNA (Cat no. 4390843) (Ambion). RAW264.7 cells (24 × 10^4^ cells/dish) were seeded in a well of 6-well plates containing 1% FCS-containing RPMI-1640 medium and cultured for 24 h. Lipofectamine^TM^ RNAiMAX reagent (Thermo Fisher Scientific) diluted with Opti-MEM™ I Reduced Serum Medium (Gibco, Waltham, MA, USA) containing the above-mentioned two types of siRNAs (each of 12.25 pmol) for mouse P2X_4_ or P2X_7_, or the negative control siRNAs (24.5 pmol) in a volume of 250 μL was applied to RAW264.7 cells. After incubation for 48 h, protein levels of P2X_4_ or P2X_7_ were determined by Western blotting, using an anti-P2X_4_ rabbit polyclonal antibody (Alomone Labs) (1:1000 dilution) and an anti-P2X_7_ rabbit polyclonal antibody (alomone labs) (1:3000 dilution) as the primary antibodies, and an HRP-conjugated anti-rabbit IgG antibody (Chemicon International) (1:5000 dilution) as the secondary antibody.

### 2.13. Determination of Phosphorylation of p38 MAP Kinase and NF-κB p65 in RAW264.7 Cells Stimulated with ATP

RAW264.7 cells (50 × 10^4^ cells/dish) were seeded in a 60-mm culture dish and cultured in 10% FCS-containing culture medium for 24 h. The cells, after incubated in 1% FCS-containing culture medium for additional 24 h, were stimulated with ATP at 1 mM for 5, 10 or 30 min and 1 or 2 h. The cells stimulated with ATP were washed with PBS and lysed in 2% SDS buffer containing 62.5 m M Tris-HCl and glycerol (pH 6.8). The collected samples were denatured at 95–100 °C for 5 min after addition of 2-mercaptoethanol and bromophenol. The protein levels of p38 MAP kinase (MAPK) and NF-κB p65, and their phosphorylated forms, i.e., *P*-p38 and *P*-p65, respectively, were determined by Western blotting as described above. The primary antibody used were: an anti-p38MAPK rabbit polyclonal antibody (Cell Signaling Technology) (1:1000 dilution), anti-phospho- p38MAPK rabbit polyclonal antibody (Cell Signaling Technol.) (1:500 dilution), anti-NF-κB p65 mouse monoclonal antibody (Santa Cruz Biotechnol., Inc., Santa Cruz, CA, USA) (1:1000 dilution) and anti-phospho-NF-κB p65 rabbit polyclonal antibody (Cell Signaling Technol.) (1:500 dilution). An HRP-conjugated anti-rabbit IgG antibody (Chemicon International, Billerica, MA, USA) (1:5000 dilution) and HRP-conjugated anti-mouse IgG antibody (Cell Signaling Technology) (1:2000 dilution) were used as secondary antibodies.

### 2.14. Determination of ATP-Induced ROS Accumulation in RAW264.7 Cells

RAW264.7 cells (2000 cells/well) were seeded in a 96-well plate and cultured in 10% FCS-containing medium for 24 h. After additional 24-h incubation in 1% FCS-containing medium, the cells were stimulated with 1 mM ATP for 30 min. After the removal of the medium, 100 μL of a ROS detection reagent solution (H_2_DFFDA, Molecular Probes Inc., Eugene, OR, USA) diluted with warmed PBS at 1 μM was added to the residual cells. After incubation for 1 h at 37 °C, the intensity of fluorescence (excitation wavelength: 485 nm, emission wavelength: 535 nm) was assessed by a fluorescence plate reader.

### 2.15. Migration Assay

A migration assay was performed using Trans-well^®^ system (24-well plate, inserts with 8-μm membrane pore size) (Corning, NY, USA) or Corning^®^ BioCoat™ Matrigel^®^ Invasion Chamber (24-well plate, inserts with 8-μm membrane pore size) (Corning), according to the manufacturer’s protocol. RAW264.7 cells (5 × 10^4^ cells/well) were seeded onto the insert and cultured in 10% FCS-containing culture medium for 24 h. After changing the medium to FCS-free medium, the cells were stimulated with ATP at 0.1 or 0.5 mM that was added to the well (lower chamber). After incubation for 3 h, the cells in the upper side of insert were wiped off with a cotton swab, and the cells migrating to the outside of the insert were fixed with 4% PFA for 10 min and stained with 1% crystal violet (FUJIFILM Wako Pure Chemical Co.) in 20% methanol for 3 min. After washing with PBS, the dried membrane was removed from the insert, and placed on a glass slide and enclosed, and the number of migrating cells was quantified by counting the cells in 5 randomly chosen visual fields.

### 2.16. Statistics Analysis

Data are represented as the mean ± SEM. Statistical significance for parametric data was analyzed by the Student’s *t*-test for two-group data and an analysis of variance followed by the Tukey’s test for multiple comparisons. For non-parametric analyses, the Kruskal-Wallis H test followed by a least significant difference-type test was employed for multiple comparisons. Significance was set at a level of *p* < 0.05.

## 3. Results

### 3.1. Novel T-Type Ca^2+^ Channel Blockers and Genetic Deletion of Ca_v_3.2 Suppress CPA-Induced Cystitis-Related Bladder Pain in Mice

CPA at 400 mg/kg caused bladder pain-like nociceptive behavior accompanied by referred hyperalgesia/allodynia in the lower abdomen and bladder swelling, as indicated by increased wet tissue weight of the bladder, 3.5–4 h after the i.p. administration in ddY mice (Figure 1A–F), in agreement with the previous reports [2,3]. We recently identified and developed a hop component, 6-PNG, and its synthetic derivative, KTt-45, as T-type Ca^2+^ channel blockers [15] when administered i.p. at 10 and 30 mg/kg, 3 h 15 min after CPA treatment. This component suppressed the CPA-induced bladder pain symptoms in a dose-dependent manner (Figure 1A,B,D,E), but not bladder swelling (Figure 1C,F). The same dose of CPA caused bladder pain symptoms (Figure 1J,K) and bladder swelling (Figure 1L) in wild-type C57BL/6J mice, and genetic deletion of Ca_v_3.2 (Figure 1J–L), as well as TTA-A2, a well-known selective T-type Ca^2+^ channel blocker (Figure 1G–I), strongly suppressed the CPA-induced bladder pain symptoms, but not bladder swelling, providing evidence for the critical role of Ca_v_3.2 in bladder pain signaling, in agreement with the previous study employing antisense oligodeoxynucleotides for Ca_v_3.2 knockdown [3].

### 3.2. CPA-Induced Over-Expression of CSE, an Enzyme Responsible for Generation of H_2_S Capable of Enhancing Ca_v_3.2 Activity, Occurs in the Mucosal Layer of the Bladder in Mice

CPA treatment significantly increased protein levels of CSE, which generates H_2_S capable of enhancing Ca_v_3.2 activity [23,24,25], in the bladder tissue homogenate (Figure 2A), which was suppressed by mesna, an acrolein-quenching drug, at a total dose of 240 mg/kg (Figure 2B), and i.p. preadministration of PPG, a CSE inhibitor, at 50 mg/kg prevented CPA-induced bladder pain-like nociceptive behavior in ddY mice (Figure 2C), as reported elsewhere [3,12]. It was also confirmed that mesna significantly reduced the CPA-induced bladder pain-like nociceptive behavior (Figure 2D) and referred hyperalgesia (Appendix A). Immuno-histochemical analyses showed that CSE was localized in both mucosal and smooth muscle layers of the bladder in the vehicle-treated control mice (Figure 2E), and CPA treatment caused serious mucosal edema and shedding of surface umbrella cells in the bladder, which clearly increased CSE staining in the mucosal layer of the bladder (Figure 2E).

### 3.3. Inhibition of the HMGB1/RAGE Pathway Reduces the CPA-Induced CSE Over-Expression and Bladder Pain Symptoms in Mice

To test whether the HMGB1/RAGE pathway [2] would be upstream of the CSE/H_2_S/Ca_v_3.2 pathway [3], we examined the effects of an anti-HMGB1-neutralizing antibody, thrombomodulin alfa capable of inactivating HMGB1 and FPS-ZM1, a RAGE antagonist, on the CPA-induced CSE upregulation in mice. Systemic preadministration of the anti-HMGB1-neutralizing antibody at 1 mg/kg, thrombomodulin alfa at 10 mg/kg or FPS-ZM1 at 1 mg/kg significantly prevented the CPA-induced CSE upregulation in the bladder tissue (Figure 3A–C) and bladder pain-like nociceptive behavior (Figure 3D–F), as well as referred hyperalgesia in the lower abdomen, but not bladder swelling (Appendix A).

### 3.4. Involvement of ATP and ROS in the CPA-Induced CSE Over-Expression and Bladder Pain Symptoms in Mice

Given the presence of evidence that CPA causes ATP-dependent macrophage infiltration and ROS generation in the bladder [13,14], we tested whether antagonists of purinergic P2X_7_ and P2X_4_ receptors and N-acetylcysteine, an antioxidant, would prevent the CPA-induced CSE over-expression and bladder pain symptoms. Systemic pre-administration of 5-BDBD, a P2X_4_ antagonist, at 3 mg/kg, but not A438079, a P2X_7_ antagonist, at 17 mg/kg, significantly attenuated the CPA-induced upregulation of CSE in the bladder (Figure 4A,B), and their combined administration strongly (synergistically) suppressed the CSE over-expression (Figure 4C). Similarly, combined administration of AZ10606120 at 1 mg/kg and PSB-12062 at 3 mg/kg, chemically unrelated selective antagonists of P2X_7_ and P2X_4_, respectively, significantly suppressed the CPA-induced CSE upregulation, although each of them failed to exhibit significant inhibitory effects (Figure 4D). N-acetylcysteine, an antioxidant, pre-administered at 100 mg/kg, significantly reduced the CPA-induced CSE over-expression in the bladder (Figure 4E). On the other hands, A438079 at 17 mg/kg, 5-BDBD at 3 mg/kg, AZ10606120 at 1 mg/kg, PSB-12062 at 1 and 3 mg/kg, and combination of A438079 and 5-BDBD or of AZ10606120 and PSB-12062 alleviated the CPA-induced bladder pain symptoms (Figure 4F–K, and Appendix A). Interestingly, the combined administration of A438079 and 5-BDBD or of AZ10606120 and PSB-12062, but not each of them, significantly reduced the CPA-induced bladder swelling (Figure 4F–K). Systemic administration of N-acetylcysteine at 100 mg/kg caused significant inhibition of the nociceptive behavior, and partial or slight inhibition of the referred hyperalgesia and bladder swelling in CPA-treated mice (Figure 4E,L).

### 3.5. Involvement of Macrophages in the CPA-Induced Bladder Pain Symptoms and CSE Upregulation in Mice

There is evidence that macrophages accumulate in the bladder mucosa of mice treated with CPA [13], macrophage-derived HMGB1 is involved in chemotherapy-induced peripheral neuropathy [8] and pancreatitis-related pain [10]. On this basis, we asked if macrophage-derived HMGB1 would contribute to the CPA-induced bladder pain symptoms and CSE over-expression in mice. CPA treatment indeed produced macrophage accumulation in the bladder mucosa (Figure 5A,B), and macrophage depletion with liposomal clodronate (Figure 5C,D) inhibited the development of bladder pain symptoms, but not swelling, following CPA treatment (Figure 5E–G). Similarly, the CPA-induced bladder pain symptoms, but not swelling, were prevented by minocycline, a macrophage/microglia inhibitor, given i.p. twice at 30 mg/kg (Appendix A), and by ethyl pyruvate, known to inhibit HMGB1 release from macrophages [26], given i.p. at 80 mg/kg (Figure 5H–J). Interestingly, the inhibition of HMGB1 release from macrophages by liposomal clodronate or ethyl pyruvate significantly reduced the CPA-induced CSE over-expression in the bladder (Figure 6).

### 3.6. ATP Release from Urothelial T24 Cells in Response to Acrolein, a Hepatic Metabolite of CPA, and ATP-Induced HMGB1 Release from Macrophage-Like RAW264.7 Cells

We next asked if acrolein, a hepatic metabolite of CPA, could cause release of ATP and/or HMGB1, DAMPs, from human urothelial T24 cells, considering the previous evidence that acrolein accumulates in the bladder and injures urothelial umbrella cells after CPA treatment [27]. Stimulation with acrolein at 50 μM for 30 min caused prompt release of ATP from T24 cells (Figure 7A). In contrast, stimulation with acrolein at 200 μM for 3 h failed to release HMGB1 from T24 cells, although acrolein at 50–200 μM caused delayed HMGB1 release in 24 h (Appendix A). Therefore, acrolein appears to cause prompt (30 min) ATP release and delayed (24 h) HMGB1 release. In RAW264.7 macrophages, stimulation with ATP at 1 mM caused HMGB1 release in 3–6 h (Figure 7B), and rapid ROS accumulation in 30 min (Figure 7C). The HMGB1 release from RAW264.7 cells in response to 3-h stimulation with 1 mM ATP was significantly suppressed by A438079 at 50 μM and 5-BDBD at 0.1 μM, P2X_7_, and P2X_4_ antagonists, respectively (Figure 7D,E). Moreover, the ATP-induced HMGB1 release from RAW264.7 cells was significantly reduced by the P2X_7_ knockdown with siRNAs (Figure 7F,H), in agreement with the inhibitory effect of A438079 (see Figure 7D). On the other hand, the P2X_4_ knockdown with siRNAs did not affect the ATP-induced HMGB1 release (Figure 7G,I), being inconsistent with the inhibitory effect of 5-BDBD (see Figure 7E). We then examined possible involvement of p38MAPK, NF-κB and ROS in ATP-induced HMGB1 release from macrophages, which were responsible for HMGB1 release in response to lipopolysaccharide [28,29] or paclitaxel, an anticancer agent [8]. SB203580 at 1 μM, a p38MAPK inhibitor, PDTC at 10 μM, an NF-κB inhibitor, and N-acetylcysteine at 50 mM, an antioxidant, significantly alleviated the ATP-induced HMGB1 release from RAW264.7 macrophages for 3 h (Figure 7J–L). ATP at 1 mM indeed caused remarkable and significant increases in phosphorylation of p38MAPK at 30 min (Figure 7M,O) and of NF-κB p65 at 2 h (Figure 7N,P). These data indicate that acrolein induces rapid ATP release from urothelial cells, and ATP induces HMGB1 through activation of p38MAPK and NF-κB and ROS accumulation following stimulation of P2X_7_ and P2X_4_ receptors.

### 3.7. Rapid Infiltration/Migration of Macrophages in Response to Relatively Low Concentrations of ATP

Given the macrophage accumulation in the bladder mucosa of mice treated with CPA (see Figure 5A,B), we tested whether ATP would initiate infiltration/migration of RAW264.7 cells in the trans-well assay system. Interestingly, stimulation with ATP even at 0.1 mM for 3 h accelerated macrophage infiltration/migration (Figure 8A), and significantly increased the number of cells that migrated to the outside of the trans-well insert (Figure 8B). We, then, tested the effects of siRNAs for P2X_7_ or P2X_4_, and found that P2X_4_ knockdown as well as P2X_7_ knockdown with siRNAs blocked the ATP-induced infiltration/migration of RAW264.7 cells (Figure 8C,D).

## 4. Discussion

Our findings from the experiments employing novel T-type Ca^2+^ channel blockers, 6-PNG and KTt-45, Ca_v_3.2-knockout mice and a CSE inhibitor provide ultimate evidence for the role of the CSE/H_2_S/Ca_v_3.2 axis in cystitis-related bladder pain, and ascertain that Ca_v_3.2 is a promising therapeutic target for treatment of bladder pain in IC/BPS patients. Our data from the in vivo inhibition experiments demonstrate that the HMGB1/RAGE axis, ATP-induced activation of both P2X_4_ and P2X_7_ receptors, ROS generation and macrophage accumulation in the bladder are upstream of the CPA-induced, acrolein-dependent over-expression of CSE in the bladder mucosa involved in the development of bladder pain. Further, the cell culture experiments employing pharmacological and gene silencing techniques show that acrolein, a hepatic metabolite of CPA, causes prompt (within 30 min) ATP release from urothelial cells, and that the released ATP triggers macrophage migration through activation of both P2X_4_ and P2X_7_ receptors and induces the release of HMGB1 from macrophages via ROS generation and activation of p38MAPK and NF-κB signals mainly via activation of P2X_7_ receptors. It is also to be noted that NF-κB is one of well-known downstream signals of RAGE targeted by HMGB1 [4,5], and involved in CPA-induced CSE over-expression in the bladder [12]. Collectively, we propose hypothetical mechanisms of cystitis-related bladder pain in CPA-treated mice, which involve a neuroimmune crosstalk mediated by ATP and HMGB1 (Figure 9). Our study for cell imaging with an H_2_S-sensing probe in the primary culture of murine urothelial cells is now in progress.

The present study demonstrates that macrophages accumulate in the bladder mucosa after CPA treatment, in agreement with the previous report [13]. We also found that the accumulating macrophages participate in the CPA-induced bladder pain, but not swelling (see Figure 5 and Appendix A), which is consistent with the findings that HMGB1 inactivation and RAGE blockade prevents the CPA-induced bladder pain without affecting bladder swelling (see Figure 3 and Appendix A). Therefore, macrophage-derived HMGB1 appears to function as a key messenger for the neuro-immune crosstalk, essential for the development of CPA-induced bladder pain, rather than inflammation.

It has been reported that bladder distention and inflammatory mediators including bradykinin, histamine and serotonin cause ATP release from urothelial cells [30], and that acrolein, a hepatic metabolite of CPA, accumulates in the bladder and injures urothelial umbrella cells after CPA treatment [27]. In the present study, acrolein induced prompt (within 30 min) ATP release and delayed (in 24 h) HMGB1 release from urothelial T24 cells (see Figure 7A, and Appendix A, respectively). Since i.p. administration of CPA causes bladder pain symptoms in 3.5–4h in vivo, we consider that ATP, released rapidly from the urothelium, in response to acrolein, might initiate the neuroimmune crosstalk, in which ATP at relatively low concentrations around 0.1 mM triggers macrophage infiltration and migration into the bladder mucosa in a manner dependent on both P2X_4_ and P2X_7_ receptors (see Figure 8). Here, ATP at relatively high concentrations around 1 mM stimulates HMGB1 release from macrophages mainly through P2X_7_ receptors (see Figure 7B,D,E). Therefore, it is likely that P2X_4_ mainly plays a role in macrophage infiltration and migration in the bladder mucosa, and P2X_7_ contributes to HMGB1 release from the accumulating macrophages adjacent to the injured urothelial cells, in addition to macrophage infiltration/migration. Nonetheless, there is evidence that P2X_4_ receptors mediate ATP-induced secretion of CXCL5 and prostaglandin E_2_ by macrophages [31,32]. The role of P2X_4_ in ATP-induced HMGB1 release from macrophages, if any, might be less important than P2X_7_, whereas P2X_4_ in addition to P2X_7_ is required for ATP-induced macrophage infiltration/migration. The cooperative functions of P2X_4_ and P2X_7_ in macrophages might interpret the synergistic inhibitory effects of P2X_4_ and P2X_7_ antagonists on the CPA-induced CSE over-expression in the bladder (see Figure 4A–D). The possibility cannot be ruled out that P2X_4_ and/or P2X_7_ expressed in non-macrophage cells might be involved in the CPA-induced CSE upregulation, cystitis and bladder pain, because the expression of, P2X_4_ and P2X_7_ in neurons and of P2X_4_ in the bladder lamina propria has been reported [33,34,35].

The ROS generation and related activation of p38MAPK and NF-κB signals are considered responsible for ATP-induced HMGB1 release from macrophages (see Figure 7J–P), being identical to the molecular mechanisms for the release of HMGB1 from macrophages in response to paclitaxel, an anti-cancer agent [8]. It is well-known that a number of cell signals, including p38MAPK and NF-κB, are activated following oxidative stress [36]. In particular, the activation of NF-κB mediates the upregulation of histone acetyltransferases (HATs), particularly CBP and PCAF, which in turn directly acetylate nuclear HMGB1 and trigger its cytoplasmic translocation followed by extracellular release [28,37,38]. We have shown that paclitaxel, an anti-cancer drug, directly causes ROS accumulation and activation of p38MAPK and NF-κB in macrophages, which causes HAT upregulation followed by cytoplasmic translocation and extracellular release of HMGB1 [8]. We consider that the ATP-induced HMGB1 might mimic the signaling pathway for paclitaxel-induced HMGB1 release from macrophages. Our previous study has shown that RAGE, but not TLR4 or CXCR4, among pronociceptive membrane receptors that HMGB1 directly or indirectly activates, is involved in the development of CPA-induced bladder pain in mice [2], which is supported by the present finding that FPS-ZM1, a selective RAGE antagonist, prevented the CPA-induced CSE over-expression in the bladder (see Figure 3C,F). The possibility that direct actions of HMGB1 on nociceptors projecting to the bladder mucosa might contribute to the CPA-induced bladder pain, cannot be ruled out and is still open to question, considering the report showing expression of functional HMGB1 receptors including RAGE in C-fiber nociceptors [39].

In summary, the present study suggests that ATP released from the urothelium stimulated with acrolein, a hepatic metabolite of CPA, initiates a neuroimmune crosstalk by triggering macrophage infiltration/migration and releasing HMGB1 from macrophages. The study also suggests that the HMGB1-induced, RAGE-dependent CSE upregulation leads to increased H_2_S generation and consequent facilitation of Ca_v_3.2 channel activity, contributing to the CPA-induced bladder pain. Pharmacological intervention, targeting ATP, P2X_4_/P2X_7_, ROS, NF-κB, HMGB1, RAGE, CSE, H_2_S or Ca_v_3.2, might serve as novel therapeutic strategies for treatment of bladder pain in IC/BPS patients.

## Figures and Tables

**Figure 1 cells-09-01748-f001:**
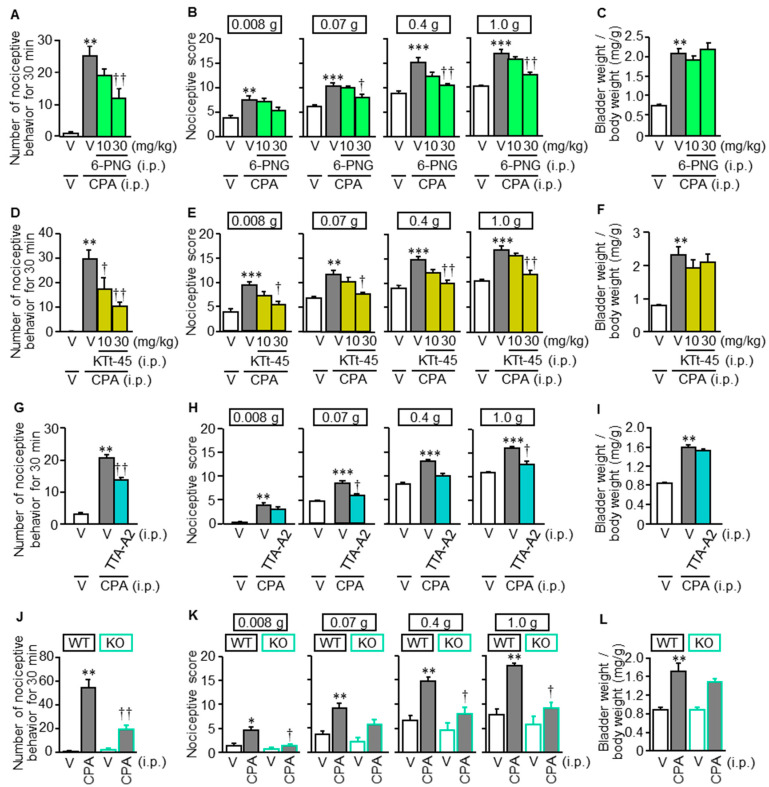
Effect of T-type Ca^2+^ channel blockers and genetic deletion of Ca_v_3.2 on CPA-induced bladder pain-like nociceptive behavior (**A**,**D**,**G**,**J**), referred hyperalgesia (**B**,**E**,**H**,**K**) and bladder swelling (**C**,**F**,**I**,**L**) in mice. (**A**–**F**) In ddY mice, 6-prenylnaringenin (6-PNG), a hop-derived T-type Ca^2+^ channel blocker, or KTt-45, a derivative of 6-PNG, at 10 and 30 mg/kg was administered i.p. 3 h 15 min after i.p. CPA at 400 mg/kg. (**G**–**I**) In C57BL/6 mice, TTA-A2 at 1 mg/kg was administered i.p. 3 h after the CPA treatment. (**J**–**L**) CPA at 400 mg/kg was administered i.p. to wild-type (WT) and Ca_v_3.2-knockout mice (KO) of a C57BL/6J background. Nociceptive behaviors were counted for 30 min starting 3.5 h after CPA treatment, followed immediately by evaluation of referred hyperalgesia/allodynia and then measurement of wet tissue weight of the excised bladder. V, vehicle. Data show the mean with S.E.M. for 6–8 (A–C), 5 (D–F), 5–6 (G–I) or 6 (J–L) mice. * *p* < 0.05, ** *p* < 0.01, *** *p* < 0.001 vs. V + V or V in WT. † *p* < 0.05, †† *p* < 0.01 vs. V + CPA or CPA in WT.

**Figure 2 cells-09-01748-f002:**
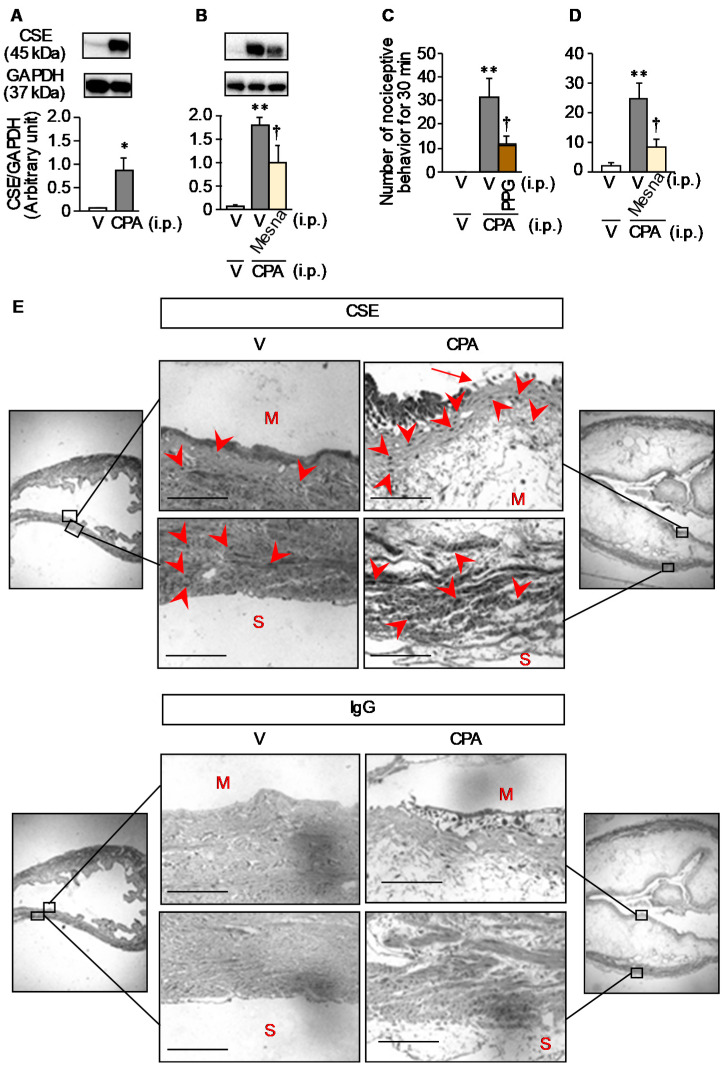
Involvement of CSE upregulation in the bladder mucosa in CPA-induced bladder pain in mice. CPA at 400 mg/kg was administered i.p. to ddY mice (**A**–**E**). Mesna, an acrolein-quenching drug, was administered orally twice, at 80 mg/kg, 30 min before, and at 160 mg/kg, 2 h after i.p. CPA. (**B**,**D**). DL-propargylglycine (PPG) at 50 mg/kg was administered i.p. 1 h before CPA treatment (**C**). In the immunostaining (**E**), the bladder tissue slices were incubated with the anti-CSE antibody or non-immune control IgG. M, mucosal side; S, serosal side; arrows, shedding of urothelial umbrella cells; arrow heads, CSE-immunoreactive cells; scale bars, 100 µm. V, vehicle. Data show the mean with S.E.M. for 5 (**A**) 5–6 (**B**,**D**) or 10–11 (**C**) mice. * *p* < 0.05, ** *p* < 0.01 vs. V or V + V; † *p* < 0.05 vs. V + CPA.

**Figure 3 cells-09-01748-f003:**
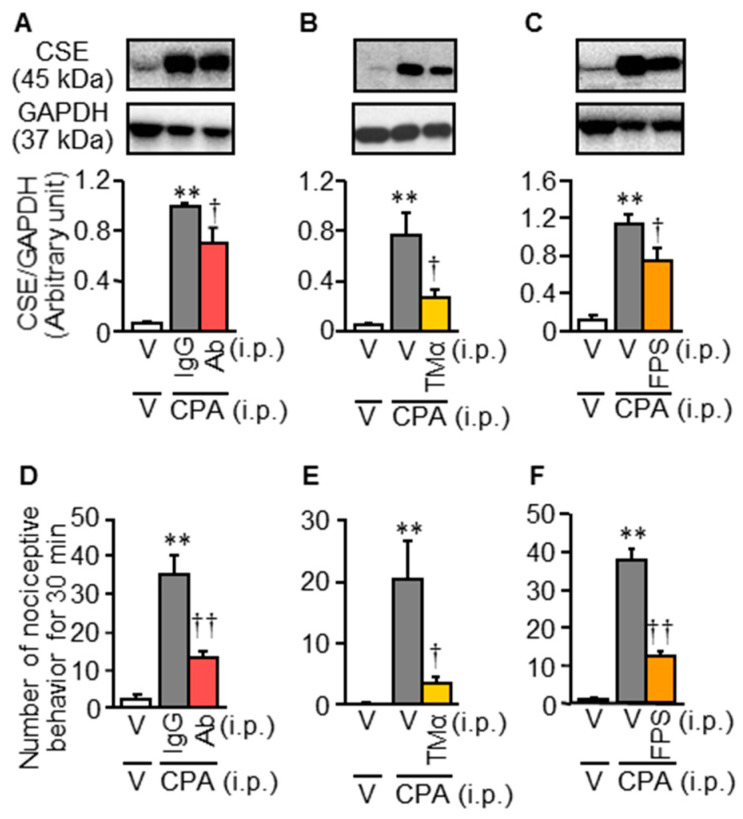
Effect of HMGB1 inactivation with an anti-HMGB1-neutralizing antibody or thrombomodulin alfa and RAGE blockade with FPS-ZM1 on the CPA-induced CSE upregulation (**A**–**C**) and bladder pain-like nociceptive behavior (**D**–**F**) in ddY mice. An anti-HMGB1-neutralizing antibody (Ab) at 1 mg/kg (**A**), thrombomodulin alfa (TMα) at 10 mg/kg or FPS-ZM1 (FPS) at 1 mg/kg was administered i.p. 30 min before i.p. CPA at 400 mg/kg. The control mice received i.p. administration of a non-immune control IgG at the same dose (**A**,**D**) or vehicle (**B**,**C**,**E**,**F**). Nociceptive behaviors were counted for 30 min starting 3.5 h after CPA treatment, and the excised bladder was subjected to Western blotting. V, vehicle. Data show the mean with S.E.M. from 5 (**A**,**B**), 6 (**C**,**F**) 5–6 (**D**,**E**) mice. ** *p* < 0.01 vs. V + V. † *p* < 0.05, †† *p* < 0.01 vs. IgG + CPA or V + CPA.

**Figure 4 cells-09-01748-f004:**
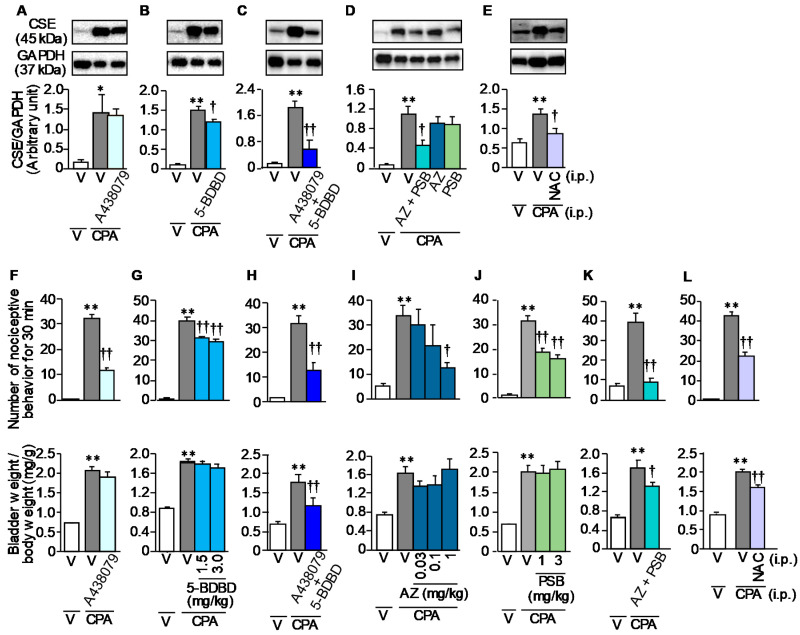
Involvement of purinergic receptors, P2X_7_ and P2X_4_, or ROS generation in CPA-induced CSE upregulation in the bladder (**A**–**E**), nociceptive behavior (top in **F**–**L**) and bladder swelling (bottom in **F**–**L**) in ddY mice. The mice received i.p. administration of A438079, a P2X_7_ antagonist, at 17 mg/kg (**A**,**F**), 5-BDBD, a P2X_4_ antagonist, at 1.5 or 3 mg/kg (**B**,**G**), A438079 at 17 mg/kg in combination with 5-BDBD at 3 mg/kg (**C**,**H**), AZ10606120 (AZ), a P2X_7_ antagonist, at 0.03, 0.1 or 1 mg/kg (**D**,**I**), PSB-12062 (PSB), a P2X_4_ antagonist, at 1 or 3 mg/kg (**D**,**J**), AZ at 1 mg/kg in combination with PSB at 3 mg/kg (**D**,**K**) or N-acetylcysteine (NAC), an antioxidant, at 100 mg/kg (**E**,**L**), 30 min before i.p. CPA at 400 mg/kg. (**A**–**E**) The protein levels of CSE in the bladder tissue assessed by Western blotting. (**F**–**L**) Nociceptive behaviors were counted for 30 min starting 3.5 h after CPA treatment, followed immediately by evaluation of referred hyperalgesia and then measurement of wet tissue weight of the excised bladder. V, vehicle. Data show the mean with S.E.M. for 5 (**A**,**C**,**E**,**G**), 4–5 (**B**), 5–6 (**D**,**F**,**K**), 7 (**H**), 5–7 (**J**), 5–10 (**I**) or 10 (**L**) mice. * *p* < 0.05, ** *p* < 0.01 vs. V + V; † *p* < 0.05, †† *p* < 0.01 vs. V + CPA.

**Figure 5 cells-09-01748-f005:**
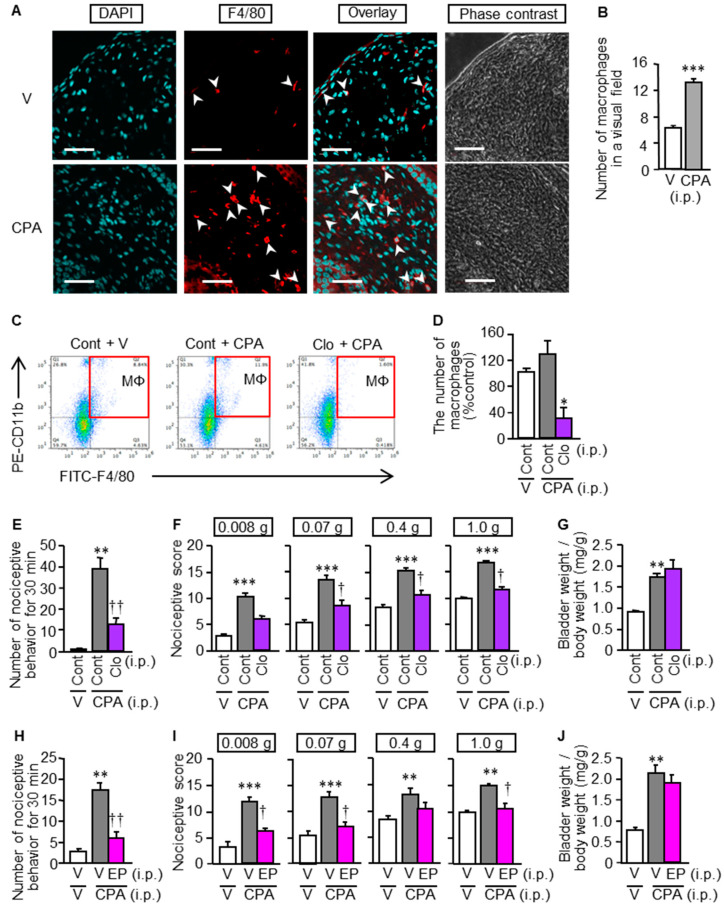
CPA-induced macrophage accumulation in the bladder (**A**,**B**) and involvement of macrophage-derived HMGB1 in CPA-induced bladder pain-like nociceptive behavior, referred hyperalgesia/allodynia and bladder swelling in ddY mice (**C**–**H**). (**A**,**B**) Accumulation of F4/80-positive macrophages in the bladder mucosa of the mice 4 h after i.p. CPA at 400 mg/kg. Typical microphotographs show accumulation of macrophages (arrow heads) stained with the anti-F4/80 antibody (red) and DAPI (nucleus, blue) in the bladder mucosa (**A**). Arrows, macrophages; scale bars, 50 µm. The number of macrophages was determined by counting F4/80-positive cells in each visual field (**B**). (**C**–**J**) Effect of macrophage depletion with liposomal clodronate (**C**–**G**) and ethyl pyruvate capable of inhibiting HMGB1 release from macrophages (**H**–**J**) on CPA-induced bladder pain and swelling in mice. Mice received i.p. administration of liposomal clodronate (Clo) or the control liposome (Cont) at 1.05 mg/mouse (**C**–**G**) and of ethyl pyruvate (EP) at 80 mg/kg (**H**–**J**), 24 h and 1 h, respectively, before i.p. CPA at 400 mg/kg. To confirm macrophage depletion with Clo, macrophages were identified (**C**) and counted (**D**) as cells positive with CD11b and F4/80 by flow cytometry. Nociceptive behaviors were counted for 30 min starting 3.5 h after CPA treatment, followed immediately by evaluation of referred hyperalgesia/allodynia and then measurement of wet tissue weight of the excised bladder. Data show the mean with S.E.M. for visual fields of 20 samples from 5 mice (**B**) and for 5–6 (**D**,**H**–**J**) or 6 (**E**–**G**) mice. * *p* < 0.05, ** *p* < 0.01, *** *p* < 0.001 vs. V + V or Cont + V; † *p* < 0.05, †† *p* < 0.01 vs. V + CPA or Cont + CPA.

**Figure 6 cells-09-01748-f006:**
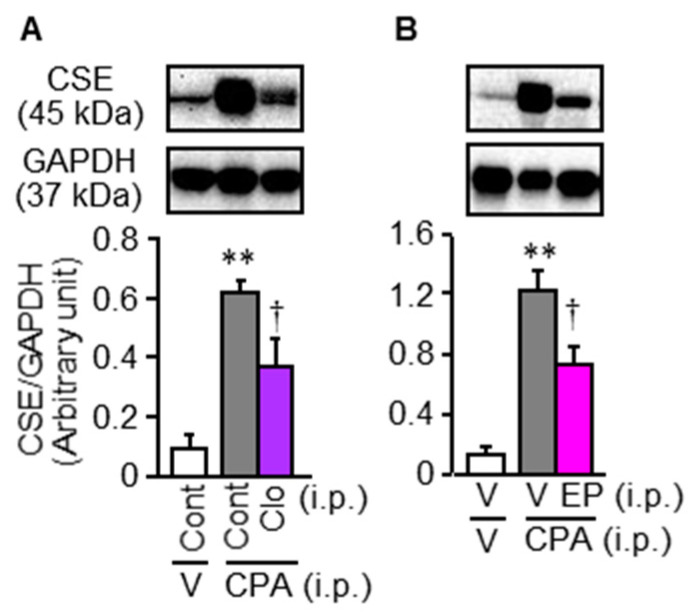
Involvement of macrophage-derived HMGB1 in CPA-induced CSE upregulation in ddY mice. Mice received i.p. administration of liposomal clodronate (Clo), a macrophage depletor, or the control liposome (Cont) at 1.05 mg/mouse (**A**) and of ethyl pyruvate (EP), capable of inhibiting HMGB1 release from macrophages, at 80 mg/kg (**B**), 24 h and 1 h, respectively, before i.p. CPA at 400 mg/kg. The bladder was excised 4 h after i.p. CPA at 400 mg/kg, and subjected to Western blotting analysis of protein levels of CSE. Data show the mean with S.E.M for 5 mice. V, vehicle. ** *p* < 0.01vs. Cont + V or V + V. † *p* < 0.05 vs. Cont + CPA or V + CPA.

**Figure 7 cells-09-01748-f007:**
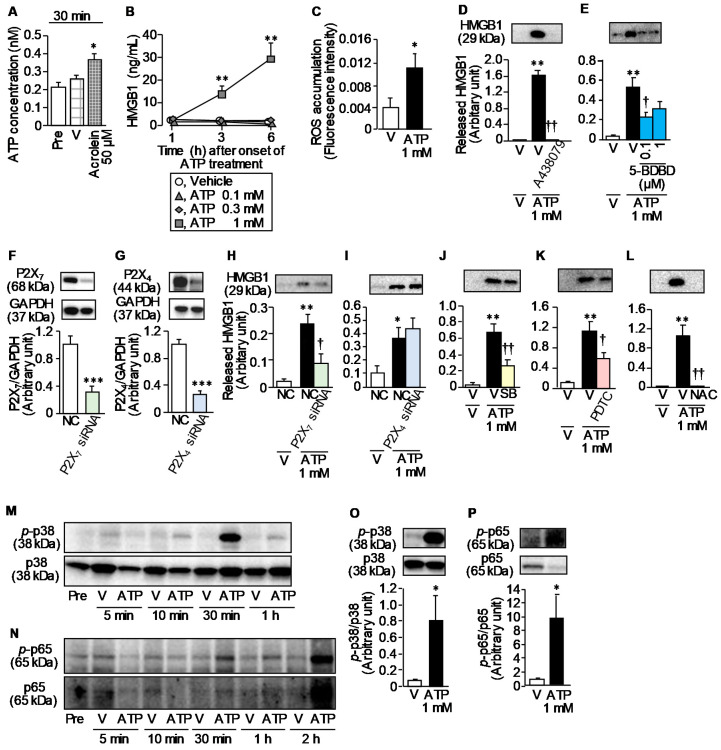
ATP release by acrolein, a hepatic metabolite of CPA, in human urothelial T24 cells and the ATP-induced HMGB1 release and ROS accumulation in murine macrophage-derived RAW264.7 cells. (**A**) Acrolein-induced prompt release of ATP from urothelial cells. T24 cells were stimulated with acrolein at 50 µM or vehicle (V) for 30 min, and the released ATP in the supernatant was determined by luciferin-luciferase reaction. (**B**) ATP-induced HMGB1 release from macrophages. RAW264.7 cells were stimulated with ATP at 0.1, 0.3 and 1 mM for 1, 3 or 6 h, and the released HMGB1 in the supernatant was determined by ELISA. (**C**) ATP-induced ROS accumulation in macrophages. RAW 264.7 cells were stimulated with ATP at 1 mM for 30 min, and the levels of ROS accumulating in the cells were determined. (**D**–**I**) Effects of A438079, a P2X_7_ antagonist (**D**), 5-BDBD, a P2X_4_ antagonist (**E**) and knockdown of P2X_7_ or P2X_4_ with siRNAs (**F**–**I**) on the ATP-induced HMGB1 release from macrophages. A438079 at 50 µM (**D**) or 5-BDBD at 0.1 or 1 µM (**E**) was added 30 min before 3-h stimulation with ATP at 1 mM, while RAW264.7 cells were subjected to 48-h treatment with siRNAs for P2X_7_ or P2X_4_, or with negative control siRNAs (NC) before the ATP challenge (**F**–**I**). The levels of released HMGB1 after ATP stimulation (**D**,**E**,**H**,**I**) and of P2X_7_ or P2X_4_ proteins after treatment with siRNAs (**F**,**G**) were determined by Western blotting. (**J**–**L**) Effects of SB203580 (SB), a p38MAPK inhibitor (**J**), pyrrolidine dithiocarbamate (PDTC), an NF-κB inhibitor (**K**), and N-acetylcysteine (NAC), an antioxidant (**L**), on the ATP-induced HMGB1 release from macrophages. RAW264.7 cells were treated with A438079 at 50 µM, 5-BDBD at 0.1 or 1 µM, SB203580 at 1 µM, PDTC at 10 µM or NAC at 50 mM, 30 min before addition of ATP at 1 mM. The levels of released HMGB1 after ATP stimulation were determined by Western blotting. (**M**–**P**) ATP-induced activation p38MAPK and NF-κB in macrophages. RAW264.7 cells were stimulated with ATP at 1 mM for 5, 10 or 30 min and for 1 or 2 h (**M**,**N**). Protein levels of p38MAPK (p38) and NF-κB p65 (p65), and of their phosphorylated forms (p-p38 and p-p65, respectively) were analyzed by Western blotting. The relative protein levels, i.e., p-p38/p38 and p-p65/p65, after stimulation with ATP at 1 mM for 30 min (**O**) or 2 h (**P**) were calculated to evaluate activation of p38 and p65. V, vehicle. Data show the mean with S.E.M. for 5 (**A**,**B**,**D**,**E**,**K**,**L**), 6 (**F**), 8 (**C**), 5–6 (**H**), 4–6 (**I**), 4–5 (**J**), 7 (**G**,**O**) or 5–8 (**P**) different experiments. * *p* < 0.05, ** *p* < 0.01, *** *p* < 0.01 vs. V, V + V, NC or NC + V; † *p* < 0.05, †† *p* < 0.01 vs. V + ATP or NC + ATP.

**Figure 8 cells-09-01748-f008:**
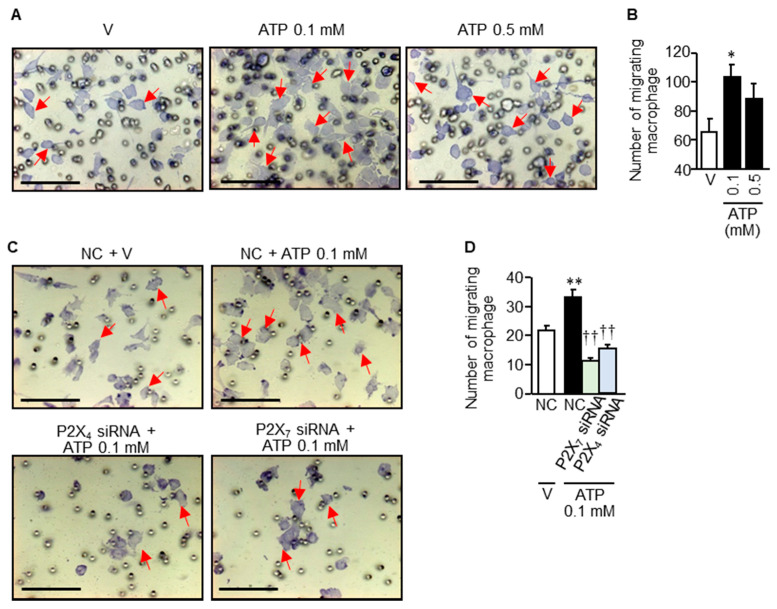
ATP-induced macrophage infiltration/migration and its inhibition by knockdown of P2X_7_ or P2X_4_ with siRNAs. (**A**,**B**) RAW264.7 cells were cultured in the trans-well insert, and stimulated with ATP at 0.1 or 0.5 mM for 3 h. (**C**,**D**) RAW264.7 cells treated with P2X_7_ siRNA, P2X_4_ siRNA or negative control siRNAs (NC) for 48 h were cultured in the trans-well insert, and stimulated with ATP at 0.1 mM for 3 h. To determine migration of the cells to the outside of the insert, the cells were counted in 5 randomly chosen visual fields. Arrows indicate typical cells that migrated to the outside of the insert. V, vehicle. Data show the mean with S.E.M. for visual fields of 20 samples from 4 different experiments (**B**), 25 samples from 5 different experiments (**D**). * *p* < 0.05, ** *p* < 0.01 vs. V or NC + V; †† *p* < 0.01 vs. NC + ATP.

**Figure 9 cells-09-01748-f009:**
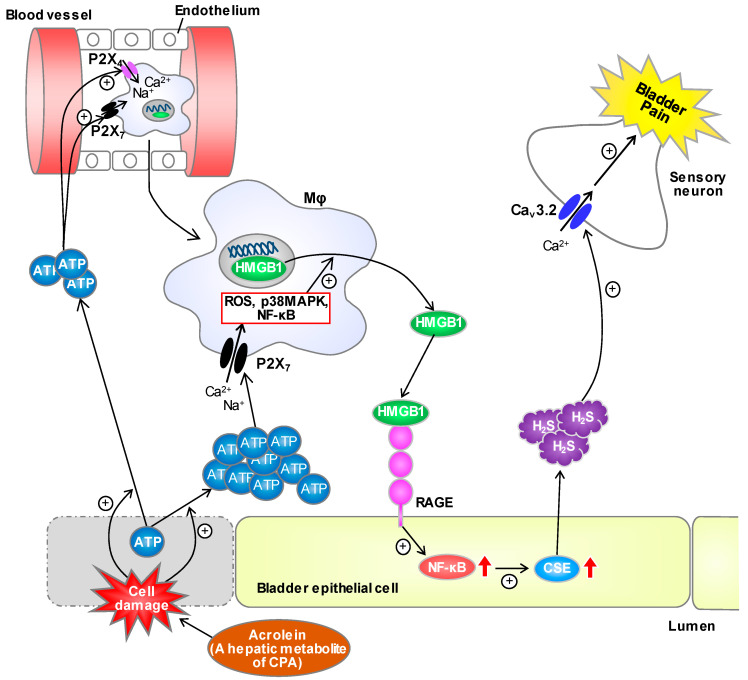
Scheme of working hypothesis for the mechanisms of a neuroimmune crosstalk responsible for CPA-induced bladder pain. ATP, once released from the urothelial cells stimulated with acrolein, a hepatic metabolite of CPA, causes macrophage infiltration and migration to the bladder mucosa via P2X_4_ and P2X_7_ and then HMGB1 release from macrophages via ROS accumulation and activation of p38MAPK and NF-κB signals mainly through P2X_7_. The released HMGB1 causes NF-κB-dependent upregulation of CSE in the urothelial cells through RAGE, and the consequently increased H_2_S generated by CSE enhances the activity of Ca_v_3.2 T-type Ca^2+^ channels expressed in the sensory neurons, resulting in bladder pain.

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
