# Peer review of "Cystitis-Related Bladder Pain Involves ATP-Dependent HMGB1 Release from Macrophages and Its Downstream H_2_S/Ca_v_3.2 Signaling in Mice"

_cells, 2020, doi:10.3390/cells9081748_

Round 1
Reviewer 1 Report
I am satisfied my original comments have been addressed in this version.
Reviewer 2 Report
The authors carefully addressed the reviewers comments and the manuscript is much improved. Therefore, I support its publication in its current form.
This manuscript is a resubmission of an earlier submission. The following is a list of the peer review reports and author responses from that submission.
Round 1
Reviewer 1 Report
Overall this is an interesting paper that investigates mechanisms of cystitis-related bladder pain. Though of interest there are some short comings regarding the context in which the work is placed, and the data interpretation based on pharmacology employed.
Introduction/Discussion
Little rationale is presented regarding why P2X receptor activation would to secretion of molecules by macrophage, particularly P2X4. Work by Layhadi et al., 2018 J Immunology outlines macrophage P2X4 activation and CXCL5 secretion, and Ulmann et al 2010 EMBO J demonstrates P2X4-dependent PGE2 secretion in macrophage. Citing this studies would help provide appropriate context/discussion points.
Results Figure 4. From the doses of A438079 and 5-BDBD used, it is difficult to ascertain selectivity and hence any conclusions based on involvement of P2X4 or P2X7. Why were these doses selected? Was a dose range employed for A438079? I gain some confidence that the effects observed are due to selective P2X7 or P2X4 inhibition, one would expect to see the effects of chemically unrelated selective antagonists (PSB-12062 for P2X4, AZ10606120 for P2X7). Alternatively KO animals would serve as a good control. Also in these studies, the investigators should discuss the roles of non-macrophage cell types mediating the effects of the antagonists e.g. neurons or epithelium.
Results Figure 7. In these reductionist experiments, there is an opportunity to use molecular knock down (siRNA) or knock out (CRISPR) in the RAW264.7 line to explore the effect of P2X4 or P2X7 depletion. This may also be useful in validating the selectivity of the receptor antagonists employed.
Reviewer 2 Report
In this paper, the authors showed that cystitis related bladder pain associated with overexpression of H2S producing enzyme cystathionine-γ-lyase protein, which results in increased Cav3.2 ion channel activity and thus bladder pain. They treated the mice with a known drug, cyclophosphamide, and showed that drug metabolite–acrolein is responsible for HMGB1 release from macrophages. The released HMGB1 binds the RAGE receptor, activates cystathionine-γ-lyase protein to produce H2S, and cause bladder pain. They showed that activation of p38MAPK and NF-kB, and elevated ROS levels are upstream events in the pathway. These results indicate that inhibition of CSE, HMGB1, p38MAPK, NF-kB, P2X4, and P2X7 are promising targets for reducing cystitis.
Minor comments:
1. Page 9, line 349-350 “Systemic preadministration of N-acetylcysteine at 100 mg/kg significantly reduced the CPA-induced bladder pain symptoms and swelling (Figure 4N-P).” but after looking at figs 4N-P, it doesn’t look like significant reduction.
2. Did authors try cyclophosphamide + 2-mercaptoethanesulfonate sodium control experiment (acrolein quenching drug used to reduce CPA side effect) to verify the acrolein role in cystitis?
3. How do elevated ROS levels in macrophages are responsible for HMGB1 expression? They should provide a reference.
4. The authors could show elevated levels of H2S after CPA treatment by doing cell imaging assay with H2S sensing probe/compound, which results in increased fluorescent intensity as compared to the control.
5. Page 16, lines 527-530, authors should delete the following sentences “Authors should discuss the results and how they can be interpreted in perspective of previous studies and of the working hypotheses. The findings and their implications should be discussed in the broadest context possible. Future research directions may also be highlighted.” from the conclusion section.